# Multi-Scale Diffusion-Guided Graph Learning with Power-Smoothing Random Walk Contrast for Multi-View Clustering

**Feiyang Chen**
College of Computer and Cyber Security
Hebei Normal University
laicc5696@gmail.com

**Ruiqiang Guo**
College of Computer and Cyber Security
Hebei Normal University
rqguo@hebtu.edu.cn

**Zhibin Gu**[*]
College of Computer and Cyber Security
Hebei Normal University
guzhibin@hebtu.edu.cn

## Abstract

Despite the notable advances in graph-based deep multi-view clustering, existing approaches still suffer from three critical limitations: (1) relying on static graph structures and being unable to model the global semantic relationships across views; (2) contamination from false negative samples in contrastive learning frameworks; and (3) a fundamental trade-off between cross-view consistency and view-specific discrimination. To address these issues, we introduce **M**ulti-sc**A**le diffusio**N**-guided **G**raph learning with p**O**wer-smoothing random walk contrast (**MANGO**) for multi-view clustering, a unified framework that combines adaptive multi-scale diffusion, random walk-driven contrastive learning, and structure-aware view consistency modeling. Specifically, the multi-scale diffusion mechanism leverages local entropy guidance to dynamically fuse similarity matrices across different diffusion steps, thereby achieving joint modeling of fine-grained local structures and overall global semantics. Additionally, we introduce a random walk-based correction strategy that explores high-probability semantic paths to filter out false negative samples, and applies a $\beta$-power transformation to adaptively reweight contrastive targets, thereby reducing noise propagation. To further reconcile the consistency-specificity dilemma, the view consistency module enforces semantic alignment across views by sharing structural embeddings, ensuring consistent local structures while preserving heterogeneous features. Extensive experiments on 12 datasets demonstrate the superior performance of MANGO.

## 1 Introduction

Multi-view Clustering (MVC) aims to partition data samples into meaningful clusters by leveraging the consensus and complementary information across multiple views. By exploiting the synergistic relationships between diverse data representations, MVC facilitates the discovery of underlying structures in complex datasets, positioning it as a key research area for integrating heterogeneous information sources and revealing intrinsic patterns (Gao et al., 2015; Liu et al., 2018; 2020; Xu et al., 2022a; Yan et al., 2024). From the perspective of learning paradigms, existing MVC methods can be broadly categorized into traditional techniques and deep learning-based models (Fang et al., 2023). Among them, deep learning-based approaches have attracted increasing attention due to their strong capacity for modeling intricate data distributions and extracting highly expressive feature representations (Huang et al., 2023; Liu et al., 2024; Tang & Liu, 2022).

Deep multi-view clustering leverages the nonlinear mapping capabilities of deep neural networks to capture the distinctive semantics of each view while effectively integrating complementary infor-

---

[*]Corresponding author

mation, enabling strong performance in complex data scenarios (Lin et al., 2023; Xu et al., 2023; Yang et al., 2023). Given the ability to explicitly model the topological relationships within the data, graph-based deep multi-view clustering (GDMVC) methods have garnered considerable attention. For example, Wen et al. (2024) proposed an adaptive hybrid graph filter that combines high- and low-frequency signals with fused multi-view embeddings to improve clustering performance on graphs. Ren et al. (2024) dynamically fuse weighted graphs using deep autoencoders and graph convolution, enabling efficient self-supervised deep multi-view clustering. Additionally, to accurately capture the affinity relationships between sample pairs, numerous contrastive learning-driven methods for graph structure refinement have been proposed Gao et al. (2024); Liu et al. (2023); Smith et al. (2025). For example, Yu et al. (2025) proposed a multi-view deep subspace clustering method leveraging contrastive learning and Cauchy-Schwarz divergence for interactive representation and clustering optimization. Chen et al. (2023) introduced a cross-view contrastive learning model that learns view-invariant and robust representations by contrasting cluster assignments across views. Additionally, Wang et al. (2023) integrated triple contrastive learning at both the feature representation and graph structure layers to generate a consensus similarity graph with a clear clustering structure.

Despite the notable progress achieved by recent graph-based deep multi-view clustering methods, three fundamental technical challenges remain unresolved. First, the reliance on static graph structures imposes inherent limitations in capturing complex semantics across multiple views. Specifically, such methods only rely on the local neighborhood relationship between samples to calculate the similarity, ignoring the global semantic connection between views. This limitation leads to inevitable information loss and distortion, which makes it difficult for the model to accurately capture complex cross-view semantic associations. Second, the issue of negative sample contamination remains prominent in graph-based contrastive learning frameworks. When constructing negative pairs, the model may mistakenly treat semantically similar samples as negatives, introducing false contrastive signals. These errors can accumulate through gradient backpropagation, forming a positive feedback loop of "noisy optimization" that progressively degrades the quality of similarity measures and weakens the effectiveness of contrastive learning. Finally, the dilemma of balancing semantic consistency and modality specificity plagues multimodal alignment strategies. This trade-off can undermine the separability of clusters, as over-alignment harms the uniqueness of the modality, while under-alignment destroys the cross-view semantic correspondence.

To address the aforementioned limitations, this study proposes Multi-scAle diffusion-guided Graph learning with pOwer-smoothing random walk contrast (MANGO) model, which contains three technical innovations. First, we introduce an adaptive multi-scale diffusion mechanism. This module dynamically fuses similarity matrices from multiple diffusion steps based on local entropy information to build a more resilient and semantically expressive graph structure. By modeling on multi-scale topology, MANGO can capture local details between directly connected samples and global semantic connections between distant samples. Second, to address the challenge of negative sample contamination, random walk path sampling is introduced to dynamically correct the sample distribution of contrastive learning. This technique explores high-probability semantic paths to filter out false negative sample pairs, and is supplemented by $\beta$-power transformation to adaptively weight negative samples. The combined method reduces noise propagation and enhances the accuracy of graph similarity estimation through iterative refinement. Third, regarding the trade-off between consistency and specificity, our structure-aware cross-view contrastive learning mechanism achieves a dual goal: to enforce semantic consistency through shared structural embeddings, while retaining modality-specific discriminative features through a view-aware attention mechanism. This balance solves the problem of cluster boundary ambiguity by coordinating global semantic alignment and local modality uniqueness. The core contributions of this paper include the following three aspects:

- A multi-scale diffusion mechanism is proposed to break the performance bottleneck brought by the fixed diffusion step size, dynamically fuse the similarity information under different step sizes, take into account both local structure exploration and global semantic modeling, and realize the effective capture of multi-granularity structural information.

- A random walk correction method is designed to optimize the distribution of contrastive learning targets. The hybrid transfer matrix is constructed by combining the t-step transfer matrix and the unit matrix, and the weight of negative samples is adjusted through $\beta$-power transformation to form a more discriminative contrast target, which reduces the impact of erroneous negative samples.

- We design a structure-aware view consistency module that simultaneously promotes semantic alignment across views and preserves modality-specific discriminative features, thereby improving clustering quality in heterogeneous multi-view scenarios.

- Extensive experiments are conducted on 12 benchmark multi-view datasets of varying types and scales. The results verify the effectiveness of MANGO compared with several state-of-the-art multi-view clustering methods.

## 2 RELATED WORK

Deep multi-view clustering (DMVC) methods can be broadly categorized into three paradigms based on how they handle inter-view relationships: joint methods, alignment-based methods, and other methods.

Joint methods integrate feature learning and clustering into unified objectives, leveraging cross-view collaboration to enhance representation quality. For instance, Li et al. (2021) jointly learned both view-specific and consensus graphs while adaptively assigning weights to obtain high-confidence clustering results. Xia et al. (2022) built a self-supervised framework based on Euler transformation and $\ell_{1,2}$-norm, integrating representation learning and clustering. Hu et al. (2023) enhanced feature-level alignment by incorporating cluster-level contrastive learning and dynamic weight learning to promote more consistent deep representations.

Alignment-based methods, in contrast, focus on mapping view-specific representations into a shared subspace to promote consistency. Early work by Hassani & Khasahmadi (2020) introduced a node-graph dual-granularity alignment framework to address cross-hierarchical redundancy. Building on this, Liu et al. (2022) mitigated representation collapse by reducing inter-view redundancy from both sample and feature perspectives. Chen et al. (2023) proposed a clustering-aware contrastive learning mechanism, which directly enforced semantic consistency across views. Trosten et al. (2023) identified negative sample bias in traditional contrastive alignment and developed a variational alignment model that maximized mutual information.

In addition, some approaches combine these strategies or address specific issues such as noise and incomplete views. For example, Luo et al. (2018) pioneered the combination of consistency constraints and view-specific modeling to establish a unified theoretical framework for multi-view subspace representation. Ke et al. (2021) built a full-process integrated framework of feature extraction-fusion-comparison-clustering, verifying the feasibility of multi-task joint optimization. Xu et al. (2021) introduced a common-specific variable dual-channel mechanism to separate multi-view shared clustering features from view-unique information.

## 3 METHOD

This section provides a detailed introduction to the proposed MANGO model, which primarily consists of four components: the self-expressive module, contrastive learning module, view consistency module, and adaptive diffusion module, as illustrated in Figure 1.

### 3.1 SELF-EXPRESSIVE MODULE

Given a multi-view dataset $\{\mathbf{X}^v \in \mathbb{R}^{n \times d_v}\}_{v=1}^m$, where $m$ is the number of views, $n$ denotes the number of samples, and $d_1, d_2, ..., d_m$ are the dimensionality of each view, we propose a self-expressive module to effectively integrate heterogeneous views. Specifically, for the $v$-th view, we first obtain its embedded representation through the encoder: $\mathbf{Z}^v = f^v(\mathbf{X}^v)$, where $f^v$ represents the encoder of $v$-th view, and $\mathbf{Z}^v$ is the learned embedded representation. The core of the module is the encoder that achieves self-reconstruction through sparse combination of latent features. The reconstruction process is defined as $\hat{\mathbf{X}}^v = \mathbf{C}^{\mathbf{v}}\mathbf{Z}^v$, where $\mathbf{C}^{\mathbf{v}}$ is the sparse coefficient matrix obtained by weighted fusion of sparse matrices under each view. For the $v-$th view, the sparse adjacency matrix within the view is generated by filtering the cosine similarity of the sample pairs with an adaptive threshold $b$, ensuring that the reconstruction process captures local structural dependencies.

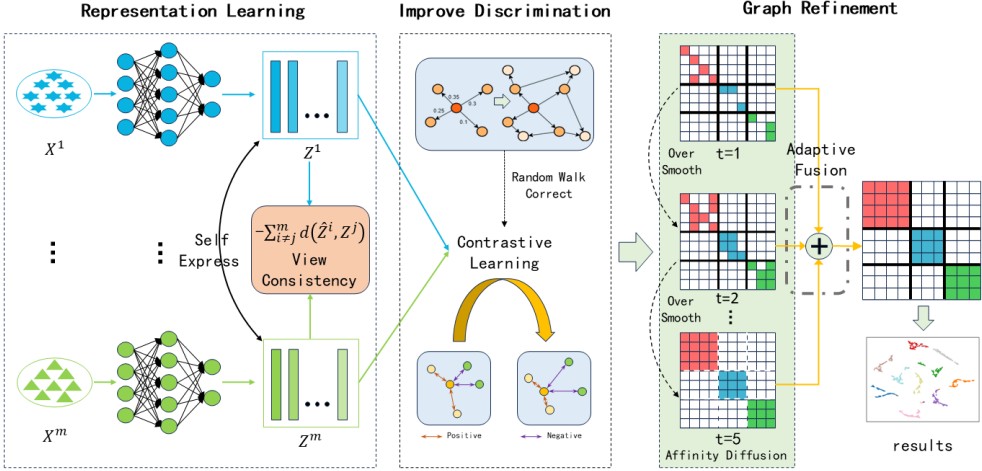

Figure 1: The Framework of MANGO. First, MANGO obtains embedding representations via view-specific MLP modules and captures data structure through reconstruction loss. Then, the view consistency module achieves semantic alignment by sharing structural embeddings, while contrastive learning and random walks filter high-probability semantic paths and eliminate pseudo-negative samples. Finally, the affinity matrix undergoes $T$ steps of diffusion, and local structures and global semantics are fused through information entropy-based weighting.

Finally, the reconstruction loss is obtained:

$$\mathcal{L}_{rec} = \frac{1}{2} \sum_{v=1}^{m} \left\| \mathbf{X}^v - \hat{\mathbf{X}}^v \right\|_F^2 \tag{1}$$

In addition, to prevent overfitting, we use the hybrid regularization term following (You et al., 2016):

$$\mathcal{L}_{reg} = \sum_{v=1}^{m} \lambda \|\mathbf{C}^v\|_1 + \frac{1-\lambda}{2} \|\mathbf{C}^v\|_F^2 \tag{2}$$

where $\lambda$ is used to balance the two regularization terms. This reconstruction loss, combined with hybrid regularization, yields embeddings that preserve both global semantics and local geometry, providing robust inputs for downstream modules.

## 3.2 POWER-SMOOTHING RANDOM WALK ENHANCED CONTRASTIVE LEARNING

Contrastive learning is widely adopted for representation learning in unlabeled multi-view scenarios, where features are aligned by contrasting positive (similar) and negative (dissimilar) pairs. However, it relies on two strong assumptions: (1) positive pairs from different views of the same sample are semantically aligned, and (2) all negative pairs are unrelated. These assumptions often fail in practice—cross-view heterogeneity can make same-class instances appear dissimilar, producing false negatives (FNs), while the absence of labels makes it difficult to ensure true negatives. Such false negatives distort training signals, disrupt the manifold structure, and weaken the discriminative power of the learned representations.

To tackle this issue, we propose a power-smoothing random walk enhanced contrastive learning strategy, which integrates two key components: a random walk correction to capture high-order semantic relations, and a power-smoothing operation to reduce the impact of false negatives by refining similarity distributions.

**Random walk-based correction mechanism:** Traditional contrastive learning assumes equal importance for all non-anchor negative samples, overlooking the intrinsic structure of the data. The proposed random walk correction mechanism simulates random walks on the sample manifold to uncover and exploit this structural information, enabling more principled weighting of negative

samples. Specifically, we first construct the affinity matrix $\mathbf{A}_{ij} = \exp(-\sigma \|\mathbf{z}_i - \mathbf{z}_j\|^2)$ through the Euclidean distance of sample embedding, where $\sigma$ is the bandwidth parameter of the Gaussian kernel. After the affinity matrix $\mathbf{A}$ is constructed, it needs to be normalized and converted into a transfer matrix $\mathbf{M}$, where $\mathbf{M}_{ij}$ represents the one-step transfer probability from sample $i$ to sample $j$. To better capture high-order manifold structures, we compute the $t$-step transition matrix $\mathbf{M}^t = \mathbf{M} \times \cdots \times \mathbf{M}$ ($t$ times), where $\mathbf{M}$ is the one-step transition matrix.

$$\mathbf{M}_{ij} = \frac{\mathbf{A}_{ij}}{\sum_{k=1}^n \mathbf{A}_{ik}} \tag{3}$$

Finally, the interpolation parameter $\eta$ is used to balance the self-connection strength and the manifold structure, formulated as $\mathbf{T} = \eta\mathbf{I} + (1 - \eta)\mathbf{M}^t$, where $\mathbf{T}$ is the target distribution matrix, and $\mathbf{T}_{ij}$ denotes the degree to which sample $j$ is a semantic neighbor of sample $i$. This value can be directly used as the negative sample weight in the intra-view contrastive loss.

**Power-smoothing-induced contrastive learning :** In addition, in order to enhance the robustness, we introduce a smoothing power operation on the basis of InfoNCE loss to control the overall strength of negative samples, which directly acts on the negative sample term in the contrast loss, thereby obtaining the expression of the intra-view contrast loss:

$$\mathcal{L}_{intra} = \frac{1}{m} \sum_{p=1}^m \left[ -\frac{1}{n} \sum_{i=1}^n \log \frac{\exp\left(\frac{s(\mathbf{z}_i^p, \mathbf{z}_i^p)}{\tau}\right)}{\exp\left(\frac{s(\mathbf{z}_i^p, \mathbf{z}_i^p)}{\tau}\right) + \left(\sum_{j \neq i} \mathbf{T}_{ij} \exp\left(\frac{s(\mathbf{z}_i^p, \mathbf{z}_j^p)}{\tau}\right)\right)^\beta} \right] \tag{4}$$

where $n$ is the number of samples, $s(\mathbf{z_i}, \mathbf{z_j})$ represents the cosine similarity, and $\beta$ is the power operation parameter. Compared with the standard InfoNCE loss, the negative sample item is subjected to the $\beta$ power operation. This operation has a nonlinear smoothing effect on the negative sample item parameter, which is used to reduce the overall impact of negative samples, especially the impact of extreme value samples.

Similarly, the expression of the contrast loss between $m$ views is as follows:

$$\mathcal{L}_{inter} = \frac{2}{m(m-1)} \sum_{p \neq q} \left[ -\frac{1}{n} \sum_{i=1}^n \log \frac{\exp\left(\frac{s(\mathbf{z}_i^p, \mathbf{z}_i^q)}{\tau}\right)}{\exp\left(\frac{s(\mathbf{z}_i^p, \mathbf{z}_i^q)}{\tau}\right) + \left(\sum_{j \neq i} \mathbf{W}_{ij} \exp\left(\frac{s(\mathbf{z}_i^p, \mathbf{z}_j^q)}{\tau}\right)\right)^\beta} \right] \tag{5}$$

The difference is that $s(\mathbf{z}_i^p, \mathbf{z}_i^q)$ represents the cosine similarity between sample $i$ in view $p$ and sample $j$ in view $q$, $\mathbf{z}_i^p$ is the embedding representation of sample $i$ in view $p$, $\mathbf{z}_j^q$ is the embedding representation of sample $j$ in view $q$, and $\mathbf{W}_{ij}$ is the uniform weight.

Finally, by integrating the aforementioned intra-view and inter-view contrastive losses, the Power-Smoothing Random Walk Enhanced Contrastive Learning framework can be formulated as follows, where $\mu$ is the balance parameter of the two contrast losses.

$$\mathcal{L}_{contra} = \mathcal{L}_{intra} + \mu\mathcal{L}_{inter} \tag{6}$$

### 3.3 VIEW CONSISTENCY MODULE

Due to potential discrepancies in noise distributions and semantic emphasis across different views, embeddings of the same class can vary significantly among views. Directly inputting such misaligned embeddings into the subsequent fusion module will destroy the intrinsic consistency of the data and fuse the conflicting noise. Therefore, in the last step of representation learning, we introduced a view consistency module, which builds a mapping bridge between views to ensure that the representations from different views can be aligned and complement each other. Specifically, view consistency is to maximize the mutual information between representations of different views. Given two view embeddings $\mathbf{Z}^p$ and $\mathbf{Z}^q$, the mutual information is defined as:

$$I\left(\mathbf{Z}^p; \mathbf{Z}^q\right) = \iint p\left(\mathbf{Z}^p, \mathbf{Z}^q\right) \log \frac{p\left(\mathbf{Z}^p, \mathbf{Z}^q\right)}{p\left(\mathbf{Z}^p\right) p\left(\mathbf{Z}^q\right)} d\mathbf{Z}^p d\mathbf{Z}^q \tag{7}$$

$$I\left(\mathbf{Z}^p; \mathbf{Z}^q\right) \geq H\left(\mathbf{h}_i\right) - \mathbb{E}_{p(\mathbf{Z}^p, \mathbf{Z}^q)}\left[d\left(f_{p \to q}\left(\mathbf{Z}^p\right), \mathbf{Z}^q\right)\right] \tag{8}$$

where $d(\cdot, \cdot)$ represents cosine distance. The core of the view consistency module is to learn a mapping function f such that: $\hat{\mathbf{Z}}^p = f_{p \to q}(\mathbf{Z}^p) \approx \mathbf{Z}^q$. Therefore, the consistency loss function can be defined as $\mathcal{L}_{p \to q} = 1 - \frac{d(\hat{\mathbf{Z}}^p, \mathbf{Z}^q)}{\tau}$, where $\tau$ is used to control the sensitivity of the loss.

For the case of $m$ views, we need to calculate the consistency loss between all view pairs:

$$\mathcal{L}_{consist} = \frac{1}{m(m-1)} \sum_{p \neq q}^{\mathbf{m}} \mathcal{L}_{p \to q} \tag{9}$$

### 3.4 Entropy-guided multi-scale diffusion for graph refinement

After obtaining discriminative representations, a graph $\mathbf{A}$ is typically constructed to encode semantic similarities among samples, serving as a foundation for downstream clustering tasks. However, most existing graph-based approaches employ static graph structures, which are highly sensitive to the quality of the learned features. To mitigate this issue, we propose an entropy-guided multi-scale diffusion strategy for graph refinement, consisting of a multi-scale diffusion module and an entropy-guided learning module.

**Multi-scale graph diffusion:** Diffusion-based methods propagate information by repeatedly multiplying the transition probability matrix, enabling global semantic aggregation. Formally, the $t$-step diffusion is $\mathbf{A}^t = \mathbf{A} \times \cdots \times \mathbf{A}$, where $\mathbf{A}$ is the sparse similarity matrix, $\mathbf{A}_{ij}$ denotes the one-step transition probability from node $i$ to $j$, and $\mathbf{A}^t$ represents the transition probability matrix after $t$ steps. Unlike traditional diffusion, we compute matrices at multiple diffusion steps to capture structural information across scales, and dynamically fuse them to balance local structures and global semantics. Specifically, given the normalized affinity matrix $\mathbf{A}_{\text{norm}}$, we construct $\{\tilde{\mathbf{A}}^0, \tilde{\mathbf{A}}^1, \ldots, \tilde{\mathbf{A}}^t\}$, where $\tilde{\mathbf{A}}^0 = \mathbf{A}_{\text{norm}}$. After each step, we retain only the top-$K$ elements per row and re-normalize to obtain $\tilde{\mathbf{A}}^t$.

**Entropy-guided multi-scale graph learning:** In this approach, entropy is used to evaluate the quality of the diffusion matrix by quantifying the uncertainty or uniformity in the distribution of connection weights. A lower entropy indicates a more concentrated distribution, which corresponds to more distinct and clearer semantic structures. Specifically, for each row $\tilde{\mathbf{A}}_i^t$ in the diffusion matrix $\tilde{\mathbf{A}}^t$, the entropy is computed over its non-zero elements, where $\tilde{\mathbf{A}}_{ij}^t$ denotes the weight of the connection between nodes $i$ and $j$ following the $t$-th diffusion step.

$$H(\tilde{\mathbf{A}}_i^t) = -\sum_{j:\tilde{\mathbf{A}}_{ij}^t > 0} \tilde{\mathbf{A}}_{ij}^t \cdot \log \tilde{\mathbf{A}}_{ij}^t \tag{10}$$

Next, the average entropy of the matrix is computed as $\bar{H}(\tilde{\mathbf{A}}^t) = \frac{1}{n} \sum_{i=1}^n H(\tilde{\mathbf{A}}_i^t)$. The inverse of entropy is used as the weight of the scale because lower entropy indicates a more concentrated distribution, corresponding to a clearer category structure. This design enables our diffusion model to automatically adjust the weights and retain complementary information at multiple scales.

$$\mathbf{A}_{fusion} = \sum_{t=0}^T \frac{1}{\bar{H}(\tilde{\mathbf{A}}^t)} \tilde{\mathbf{A}}^t \tag{11}$$

To facilitate subsequent spectral clustering, we further apply symmetric normalization and diagonal enhancement to the final diffusion matrix. Specifically, the symmetric normalization is achieved by averaging the matrix with its transpose, while diagonal enhancement is performed by scaling the diagonal elements using an enhancement coefficient $k$.

$$\mathbf{A}_{final}[i,j] = \frac{1}{2}(\mathbf{A}_{fusion}[i,j] + \mathbf{A}_{fusion}[j,i]) \cdot k \tag{12}$$

## 3.5 THE OVERALL LOSS FUNCTION

Combining self-representation and regularization losses, by jointly random Walk modified power smoothing contrastive learning and view consistency modules, the overall loss function of our proposed MANGO is formulated as follows:

$$\mathcal{L} = \mathcal{L}_{reg} + \alpha\mathcal{L}_{rec} + \beta\mathcal{L}_{contra} + \gamma\mathcal{L}_{consist} \tag{13}$$

where hyper-parameters $\alpha$, $\beta$, and $\gamma$ balance the importance of the three terms.

## 4 EXPERIMENT

### 4.1 EXPERIMENTAL SETTINGS

**Datasets:** Twelve datasets with varying types and scales are used: Yale (Chowdhury et al., 2025), ORL (Du et al., 2022), BBC-Sport (Chowdhury et al., 2025), Reuters (Du et al., 2022), Scene-15 (Fei-Fei & Perona, 2005) , MSRC-v1 (Du et al., 2022), LandUse-21 (Zong et al., 2022), Caltech101-20 (Peng et al., 2019), ALOI-100 (Yuan et al., 2025), STL10 (Yu et al., 2024), HandWritten (Van Breukelen et al., 1998), and MNIST-3V (Zhou et al., 2020). More detailed descriptions of these datasets can be found in Table 1.

**Baselines:** We compare MANGO with eight SOTA multi-view clustering methods, including **MFLVC**(2022) (Xu et al., 2022b), **MSESC**(2023) (Cui et al., 2023), **CVCL**(2023) (Chen et al., 2023), **LSGMC**(2023) (Lan et al., 2023), **MVD**(2023) (Li et al., 2023), **DIVIDE**(2024) (Lu et al., 2024), **SCM**(2024) (Luo et al., 2024), **CANDY**(2024) (Guo et al., 2024).

**Evaluation metrics:** All results are the average of 10 independent runs. We have calculated the average. To comprehensively evaluate clustering performance, we adopt three widely used metrics: clustering accuracy (ACC), normalized mutual information (NMI), and adjusted Rand index (ARI), where higher values indicate better performance.

Table 1: The detail for experimental datasets

| Dataset | Type | # Instances | # Classes | # Views |
|---|---|---|---|---|
| **Yale** | Face | 165 | 15 | 3 |
| **ORL** | Face | 400 | 40 | 3 |
| **BBC-Sport** | Text | 544 | 5 | 2 |
| **Reuters** | Text | 1200 | 6 | 5 |
| **Scene-15** | Scene | 4485 | 15 | 3 |
| **MSRC-v1** | Object | 210 | 7 | 5 |
| **LandUse-21** | Object | 2100 | 21 | 3 |
| **Caltech101-20** | Object | 2386 | 20 | 6 |
| **ALOI-100** | Object | 10800 | 100 | 4 |
| **STL10** | Object | 13000 | 10 | 3 |
| **HandWritten** | Digit | 2000 | 10 | 6 |
| **MNIST-3V** | Digit | 60000 | 10 | 3 |

For fair comparison, the hyperparameters of all baseline methods are carefully tuned based on their publicly available code, and the best-performing settings are adopted. For the MANGO model, a three-layer MLP is employed to extract features for each view, with hidden layer sizes set to 1024, 512, 256. The input dimension corresponds to the original feature size, and the output dimension is fixed at 256. In all experiments, the bandwidth parameter of the Gaussian kernel $\sigma$ is set to 0.3, the regularization parameter $\lambda$ to 0.3, the interpolation parameter $\eta$ to 1.2, the number of random walk steps $t$ to 3, the temperature parameter $\tau$ to 0.6. The contrastive loss weight $\mu$ is selected from the set {0.01, 0.1, 1, 10} through a grid search, and is ultimately set to 0.1 across all datasets. Hyperparameters $\alpha$, $\beta$, and $\gamma$ are selected via grid search over the set {1e3,1e4}, {1e4,1e5} and {1e5,1e6}. Shallow learning experiments are implemented in MATLAB 2023b on a workstation with a 2.50GHz 7285H 32-core CPU and 128 GB RAM, while deep learning experiments are conducted using PyTorch 2.5.1 on an H20-NVLink GPU.

### 4.2 COMPARISON RESULTS

Table 2 records the experimental comparison of our proposed MANGO with other 8 comparison methods across twelve datasets, where the best and suboptimal performance are highlighted in bold

Table 2: Clustering performance of all methods on twelve datasets

| Metric | Dataset | MFLVC | MSESC | CVCL | LSGMC | MVD | DIVIDE | SCM | CANDY | MANGO |
|--------|---------|-------|-------|------|-------|-----|--------|-----|-------|-------|
| | | ACC | 0.572±0.053 | 0.680±0.040 | 0.687±0.022 | 0.711±0.009 | 0.661±0.000 | 0.607±0.013 | 0.574±0.025 | 0.590±0.025 | **0.729±0.013** |
| | Yale | NMI | 0.547±0.050 | 0.660±0.028 | 0.672±0.017 | 0.724±0.009 | 0.687±0.001 | 0.652±0.019 | 0.586±0.020 | 0.600±0.029 | **0.757±0.011** |
| | | ARI | 0.375±0.042 | 0.529±0.036 | 0.442±0.019 | 0.531±0.032 | 0.418±0.002 | 0.431±0024 | 0.395±0.026 | 0.353±0.035 | **0.582±0.015** |
| | | ACC | 0.422±0.036 | 0.695±0.041 | 0.832±0.029 | 0.853±0.029 | 0.882±0.021 | 0.693±0.041 | 0.673±0.030 | 0.672±0.019 | **0.926±0.007** |
| | ORL | NMI | 0.573±0.031 | 0.728±0.028 | 0.865±0.031 | 0.938±0.006 | 0.951±0.004 | 0.831±0.024 | 0.821±0.017 | 0.623±0.008 | **0.961±0.004** |
| | | ARI | 0.272±0.030 | 0.573±0.042 | 0.585±0.035 | 0.891±0.024 | 0.842±0.015 | 0.575±0.056 | 0.558±0.036 | 0.567±0.020 | **0.895±0.011** |
| | | ACC | 0.697±0.042 | 0.720±0.066 | 0.601±0.068 | 0.936±0.000 | 0.785±0.000 | 0.447±0.015 | 0.720±0.034 | 0.639±0.066 | **0.959±0.012** |
| | BBC-Sport | NMI | 0.514±0.037 | 0.695±0.031 | 0.331±0.073 | 0.846±0.000 | 0.683±0.000 | 0.151±0.036 | 0.565±0.025 | 0.404±0.053 | **0.907±0.019** |
| | | ARI | 0.482±0.034 | 0.584±0.051 | 0.276±0.072 | 0.829±0.000 | 0.622±0.000 | 0.109±0.004 | 0.513±0.031 | 0.358±0.072 | **0.932±0.018** |
| | | ACC | 0.422±0.032 | 0.397±0.030 | 0.487±0.046 | 0.467±0.003 | 0.461±0.027 | 0.563±0.015 | 0.459±0.061 | 0.539±0.024 | **0.583±0.026** |
| | Reuters | NMI | 0.220±0.031 | 0.185±0.020 | 0.271±0.031 | 0.334±0.004 | 0.334±0.005 | **0.363±0.008** | 0.242±0.040 | 0.284±0.022 | 0.351±0.013 |
| | | ARI | 0.161±0.024 | 0.111±0.018 | 0.214±0.035 | 0.241±0.002 | 0.257±0.013 | **0.293±0.043** | 0.201±0.055 | 0.325±0.017 | 0.281±0.013 |
| | | ACC | 0.313±0.011 | 0.373±0.025 | 0.367±0.015 | 0.454±0.002 | 0.413±0.002 | 0.474±0.020 | 0.325±0.012 | 0.365±0.022 | **0.497±0.023** |
| | Scene-15 | NMI | 0.327±0.005 | 0.362±0.024 | 0.387±0.015 | 0.490±0.001 | 0.381±0.002 | 0.485±0.008 | 0.263±0.014 | 0.359±0.011 | **0.499±0.004** |
| | | ARI | 0.173±0.007 | 0.204±0.023 | 0.309±0.013 | 0.314±0.001 | 0.238±0.000 | 0.307±0.016 | 0.150±0.009 | 0.195±0.013 | **0.388±0.013** |
| | | ACC | 0.872±0.057 | 0.822±0.044 | 0.911±0.027 | 0.938±0.002 | 0.871±0.000 | 0.732±0.035 | 0.665±0.032 | 0.494±0.029 | **0.949±0.006** |
| | MSRC-v1 | NMI | 0.752±0.048 | 0.673±0.034 | 0.855±0.024 | 0.882±0.003 | 0.751±0.000 | 0.653±0.033 | 0.592±0.034 | 0.287±0.030 | **0.904±0.013** |
| | | ARI | 0.739±0.046 | 0.602±0.033 | 0.823±0.026 | 0.859±0.004 | 0.712±0.000 | 0.615±0.032 | 0.493±0.040 | 0.160±0.034 | **0.884±0.014** |
| | | ACC | 0.244±0.036 | 0.249±0.013 | 0.295±0.010 | 0.303±0.005 | 0.259±0.007 | 0.313±0.013 | 0.260±0.010 | 0.229±0.027 | **0.315±0.009** |
| | LandUse-21 | NMI | 0.245±0.044 | 0.273±0.010 | 0.293±0.009 | 0.343±0.003 | 0.331±0.003 | 0.268±0.073 | 0.291±0.010 | 0.258±0.032 | **0.344±0.003** |
| | | ARI | 0.095±0.018 | 0.104±0.008 | 0.111±0.007 | 0.157±0.003 | 0.114±0.003 | 0.174±0.079 | **0.278±0.007** | 0.091±0.019 | 0.160±0.004 |
| | | ACC | 0.613±0.031 | 0.472±0.039 | 0.506±0.043 | 0.593±0.022 | 0.573±0.013 | 0.616±0.026 | 0.518±0.058 | 0.421±0.040 | **0.619±0.023** |
| | Caltech101-20 | NMI | **0.742±0.013** | 0.641±0.027 | 0.354±0.020 | 0.722±0.010 | 0.689±0.007 | 0.627±0.010 | 0.513±0.020 | 0.393±0.237 | 0.725±0.014 |
| | | ARI | 0.522±0.033 | 0.433±0.041 | 0.472±0.041 | 0.488±0.033 | 0.491±0.012 | 0.504±0.017 | **0.627±0.095** | 0.246±0.019 | 0.535±0.032 |
| | | ACC | 0.693±0.027 | 0.557±0.015 | 0.756±0.025 | 0.547±0.009 | 0.703±0.013 | 0.753±0.018 | 0.687±0.015 | 0.696±0.027 | **0.887±0.009** |
| | ALOI-100 | NMI | 0.775±0.035 | 0.727±0.009 | 0.701±0.016 | 0.683±0.004 | 0.821±0.002 | 0.834±0.042 | 0.800±0.006 | 0.702±0.021 | **0.910±0.003** |
| | | ARI | 0.562±0.041 | 0.418±0.209 | 0.529±0.027 | 0.327±0.010 | 0.584±0.010 | **0.611±0.023** | 0.516±0.013 | 0.568±0.017 | 0.803±0.009 |
| | | ACC | 0.112±0.002 | 0.915±0.073 | 0.702±0.042 | 0.831±0.000 | 0.111±0.000 | 0.917±0.060 | 0.937±0.016 | 0.677±0.008 | **0.960±0.013** |
| | STL10 | NMI | 0.001±0.000 | 0.865±0.069 | 0.452±0.027 | 0.838±0.000 | 0.001±0.000 | 0.828±0.020 | 0.867±0.021 | 0.615±0.010 | **0.911±0.012** |
| | | ARI | 0.000±0.000 | 0.815±0.079 | 0.467±0.030 | 0.770±0.000 | 0.000±0.000 | 0.838±0.010 | 0.872±0.011 | 0.594±0.009 | **0.912±0.021** |
| | | ACC | 0.851±0.046 | 0.905±0.074 | 0.892±0.057 | 0.976±0.000 | 0.871±0.027 | 0.867±0.052 | 0.730±0.055 | 0.951±0.016 | **0.978±0.001** |
| | HandWritten | NMI | 0.849±0.032 | 0.858±0.057 | 0.865±0.054 | 0.944±0.000 | 0.913±0.022 | 0.870±0.019 | 0.697±0.030 | 0.885±0.021 | **0.948±0.001** |
| | | ARI | 0.790±0.049 | 0.827±0.080 | 0.822±0.079 | 0.947±0.000 | 0.853±0.021 | 0.817±0.042 | 0.600±0.052 | 0.842±0.032 | **0.952±0.001** |
| | | ACC | 0.982±0.010 | 0.970±0.030 | 0.897±0.000 | OM | OM | 0.984±0.008 | 0.931±0.027 | **0.994±0.002** | 0.987±0.000 |
| | MNIST-3V | NMI | 0.954±0.019 | 0.943±0.011 | 0.7803±0.001 | OM | OM | 0.954±0.000 | 0.862±0.015 | **0.964±0.003** | 0.962±0.001 |
| | | ARI | 0.960±0.022 | 0.946±0.299 | 0.7104±0.001 | OM | OM | 0.965±0.001 | 0.830±0.024 | 0.971±0.002 | **0.972±0.000** |

and underline respectively, and the abbreviation "OM" indicates the occurrence of out of memory error. Through comprehensive experiments, we have the following observations:

1) Our MANGO model shows excellent clustering performance in all datasets, significantly outperforming its competitors in some scenarios. In particular, on the ALOI-100 dataset, our MANGO achieves 88.7% ACC, which is about 13.4% higher than the second-best algorithm DIVIDE. These results suggest that MANGO effectively exploits the rich information among multi-view data by jointly leveraging the random walk-enhanced contrastive learning module and the view consistency mechanism.

2) Compared with existing contrastive learning-based algorithms (such as CANDY, DIVIDE, and SCM), the MANGO model achieves the best clustering performance in most cases. This demonstrates that our power-smoothing and random walk-enhanced contrastive learning mechanism effectively improves representation quality, which in turn leads to enhanced clustering results.

3) Shallow methods like LSGMC and MVD perform well on small datasets but struggle with scalability and often face out-of-memory errors on larger ones due to limited representational capacity. Deep methods such as SCM and CANDY improve on this by learning more expressive features, achieving better results on complex datasets like STL10 and MNIST-3V. Nonetheless, the proposed MANGO model consistently outperforms existing shallow and deep multi-view clustering methods in most cases, demonstrating its superior performance and robustness.

4) To further verify the effectiveness of our method, we take the HandWritten dataset as an example to visualize the clustering results of other MVC methods and our method. The t-SNE results are shown in Figure 2. It can be seen that our method obtains more clear and compact clusters, which further confirms the superiority of our method.

## 4.3 PARAMETER SENSITIVITY ANALYSIS

This section studies the impact of three hyper-parameters $\alpha$, $\beta$, and $\gamma$ on the MANGO model. Specifically, we perform grid search by adjusting $\alpha$, $\beta$, and $\gamma$ in the set {1e3, 3e3, 5e3, 7e3, 9e3}, {1e4,

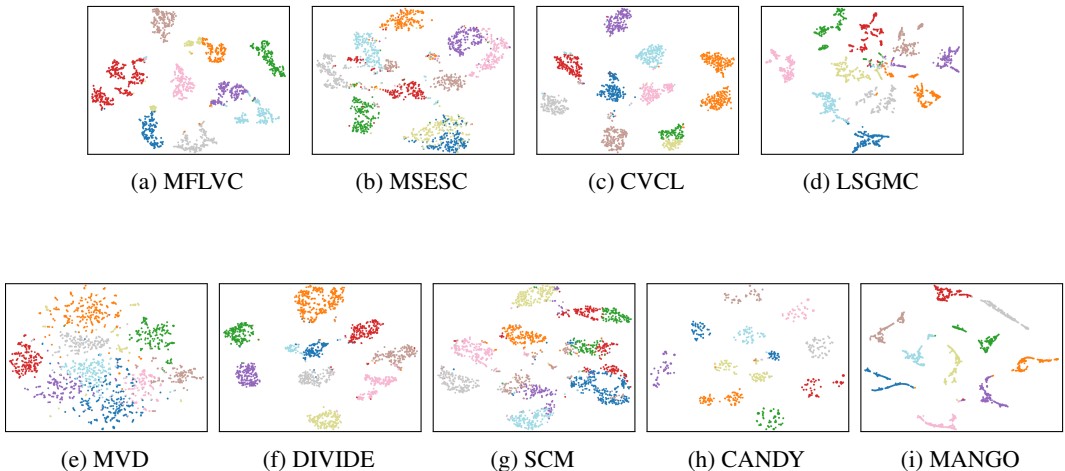

Figure 2: t-SNE visualization of the consensus affinity matrix on the HandWritten dataset

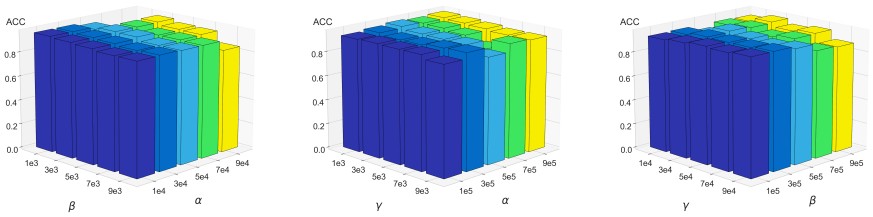

Figure 3: Parameters sensitivity analysis with parameters $\alpha$, $\beta$, and $\gamma$ on MSRC-v1.

3e4, 5e4, 7e4, 9e4}, and {1e5, 3e5, 5e5, 7e5, 9e5} respectively. Figure 3 shows how the performance of the model changes with various combinations of these parameters. The results indicate that the MANGO model performs well across the specified ranges of $\alpha$, $\beta$, and $\gamma$, demonstrating robustness to variations in these hyperparameters.

## 4.4 ABLATION STUDY

Finally, we conduct comprehensive ablation experiments to evaluate the contribution of each module within the MANGO model. Specifically, we remove the contrastive loss ($\mathcal{L}_{contrat}$), view consistency loss ($\mathcal{L}_{consist}$), false negative (FN) adjustment strategy, and adaptive diffusion module from the complete MANGO model in various combinations and record the corresponding performance. The results on the MSRC-v1 and Reuters datasets are summarized in Table 3. It is evident that the full MANGO model achieves the best performance, demonstrating that the modules work synergistically to deliver superior clustering results. Moreover, the performance of configuration (c) surpasses that of (a), and (h) outperforms (g), indicating that the random walk-enhanced contrastive learning and view consistency modules effectively exploit view consistency, while the adaptive diffusion module efficiently captures the underlying graph structure.

Furthermore, to further verify how multi-scale diffusion jointly models local and global structures, we conducted two experiments. The first was a comparison between the complete MANGO model and its variant (denoted as MANGO-w). This variant removed multi-scale fusion and used only a fixed 3-step diffusion step size. The experimental results are as follows. The Table 4 show that for large-scale data with complex structures, relying solely on structural information at a single scale is insufficient. The model must be able to simultaneously perceive subtle local relationships and macroscopic global communities, which is precisely what multi-scale diffusion provides.

Table 3: Ablation study on MSRC-v1 and Reuters dataset

| | $\mathcal{L}_{rec}$ | $\mathcal{L}_{contra}$ | $\mathcal{L}_{consist}$ | random | diffusion | MSRC-v1 | | | Reuters | | |
|---|---|---|---|---|---|---|---|---|---|---|---|
| | | | | | | ACC | NMI | ARI | ACC | NMI | ARI |
| (a) | ✓ | | | | | 0.770 | 0.758 | 0.662 | 0.502 | 0.347 | 0.267 |
| (b) | ✓ | ✓ | | | | 0.795 | 0.780 | 0.699 | 0.510 | 0.347 | 0.269 |
| (c) | ✓ | ✓ | | ✓ | | 0.800 | 0.781 | 0.700 | 0.535 | 0.364 | 0.286 |
| (d) | ✓ | ✓ | | ✓ | ✓ | 0.893 | 0.845 | 0.902 | 0.507 | 0.344 | 0.258 |
| (e) | ✓ | | ✓ | | | 0.781 | 0.749 | 0.660 | 0.542 | 0.360 | 0.279 |
| (f) | ✓ | | ✓ | | ✓ | 0.790 | 0.770 | 0.681 | 0.546 | 0.360 | 0.270 |
| (g) | ✓ | ✓ | ✓ | ✓ | | 0.863 | 0.818 | 0.748 | 0.551 | 0.340 | 0.261 |
| (h) | ✓ | ✓ | ✓ | ✓ | ✓ | **0.950** | **0.908** | **0.887** | **0.587** | **0.376** | **0.287** |

Table 4: Ablation study for multi-scale diffusion

| Dataset | Method | ACC | NMI | ARI | Dataset | Method | ACC | NMI | ARI |
|---|---|---|---|---|---|---|---|---|---|
| Yale | MANGO-w | 0.715 | 0.750 | 0.572 | ORL | MANGO-w | 0.928 | 0.961 | 0.895 |
| | MANGO | 0.729 | 0.757 | 0.582 | | MANGO | 0.926 | 0.961 | 0.895 |
| BBC-Sport | MANGO-w | 0.953 | 0.906 | 0.921 | Reuters | MANGO-w | 0.499 | 0.321 | 0.253 |
| | MANGO | 0.959 | 0.907 | 0.932 | | MANGO | 0.583 | 0.351 | 0.281 |
| Scene-15 | MANGO-w | 0.496 | 0.502 | 0.343 | MSRC-v1 | MANGO-w | 0.929 | 0.875 | 0.841 |
| | MANGO | 0.497 | 0.499 | 0.388 | | MANGO | 0.949 | 0.904 | 0.884 |
| LandUse21 | MANGO-w | 0.301 | 0.344 | 0.148 | Caltech101-20 | MANGO-w | 0.597 | 0.713 | 0.494 |
| | MANGO | 0.315 | 0.344 | 0.160 | | MANGO | 0.619 | 0.725 | 0.535 |
| ALOI-100 | MANGO-w | 0.828 | 0.878 | 0.731 | STL-10 | MANGO-w | 0.926 | 0.862 | 0.837 |
| | MANGO | 0.887 | 0.910 | 0.803 | | MANGO | 0.960 | 0.911 | 0.912 |
| HandWritten | MANGO-w | 0.973 | 0.944 | 0.944 | MNIST-3V | MANGO-w | 0.985 | 0.960 | 0.970 |
| | MANGO | 0.978 | 0.948 | 0.952 | | MANGO | 0.987 | 0.962 | 0.972 |

## 5 CONCLUSION

This paper proposes a novel deep multi-view clustering framework, which effectively learns a multi-view embedding representation with strong discriminative power by integrating the random walk modified contrastive learning module and the view consistency module. Among them, the random walk modified contrastive learning module enhances the adaptability of the model to complex data distribution by dynamically adjusting the weights of negative samples; the view consistency module realizes deep alignment across view feature spaces through a bidirectional projection mechanism. In addition, the introduction of the adaptive diffusion module can dynamically capture the multi-scale structural information of the data, effectively avoiding the over-smoothing and information loss problems commonly seen in traditional methods. Extensive experiments fully verify the superiority and effectiveness of MANGO.

## ACKNOWLEDGEMENTS

This work was supported in part by the National Natural Science Foundation of China (No. 62506116), in part by the Hebei Natural Science Foundation (No. F2025205006), and in part by the Science Research Project of the Hebei Education Department (No. BJ2026004).

## ETHICS STATEMENT

In this study, we propose a novel deep multi-view clustering framework to enhance its representation learning capabilities.This research did not involve human subjects, human-related data (e.g., personal identifiers, behavioral records), or animal subjects. This research did not receive any external sponsorship or funding, and none of the authors have any financial, professional, or personal conflicts of interest. Throughout this research, we strictly adhered to the principles of research integrity. All experimental procedures, data analysis, and result interpretation were performed in an objective and transparent manner, with full records maintained for verification. No ethical violations, such as data fabrication, manipulation, or plagiarism, occurred at any stage.

## REPRODUCIBILITY STATEMENT

To ensure the reproducibility of our work, we have uploaded the source code. All datasets used in our experiments are from public datasets. In addition, all experimental procedures and result reports follow transparent standards. Section 4.1 of the main paper details the evaluation metrics (e.g., NMI, ACC, ARI) and includes the hyperparameter search range. To account for the randomness of model initialization and data partitioning, we report the average results of 10 independent runs.

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

# A    TECHNICAL APPENDICES AND SUPPLEMENTARY MATERIAL

## A.1    ALGORITHM

The entire algorithm of MANGO is summarized in Algorithm 1

---

**Algorithm 1** The algorithm of MANGO

---

1: **Input**: Multi-view data $\{\mathbf{X}^v \in \mathbb{R}^{n \times d_v}\}_{v=1}^m$; Training iterations $E$; Trade-off coefficients $\alpha$; $\beta$ and $\gamma$; diffusion steps $T$.
2: **Output $\mathbf{A}_{final}$.**
3: **for** $epoch = 1$ to $E$ **do**
4:     Compute embeddings $\{\mathbf{Z}^v\}_{v=1}^m$ via $\{MLP^v\}_{v=1}^m$.
5:     Compute reconstruction loss$\mathcal{L}_{rec}$ through Eq. 1.
6:     Compute regularization loss$\mathcal{L}_{reg}$ through Eq. 2.
7:     Compute contrastive loss$\mathcal{L}_{inter}$ through Eq. 5.
8:     Compute consistency loss$\mathcal{L}_{consist}$ through Eq. 9.
9:     Update the network by optimizing $\mathcal{L}$ in Eq. 13.
10: **end for**
11: Build affinity matrix $\mathbf{A}$.
12: **for** $t = 1$ to $T$ **do**
13:     Compute the information entropy of $\tilde{\mathbf{A}}^t$ through Eq. 10.
14: **end for**
15: Compute final affinity matrix$\mathbf{A}_{final}$ through Eq. 12
16: Performing the spectral clustering on $\mathbf{A}_{final}$ to obtain the final clustering results.

---

## A.2    ADDITIONAL EXPERIMENTAL RESULTS

In this section, we provide complete results on all datasets, including parameter sensitivity analysis and ablation experiments. Table 5-Table 10 summarize the ablation studies of MANGO on all datasets for the three loss items and other improvements, while Figure 4 shows the sensitivity of MANGO to parameters $\alpha$, $\beta$, and $\gamma$ on all datasets.

As shown in Table 5 to Table 10, the complete MANGO model consistently outperforms its ablated variants across all datasets. This demonstrates that the integration of the self-expressive module, contrastive learning module, view consistency module, and adaptive diffusion module enables MANGO to fully exploit the rich information embedded in multi-view data, thereby enhancing clustering performance.

As for the parameter sensitivity analysis, Figure 4 demonstrates that MANGO consistently achieves stable and accurate clustering results across all 12 datasets over a broad range of parameter values, highlighting its robustness and practical reliability.

Table 5: Ablation study on Yale and ORL dataset

| | $\mathcal{L}_{rec}$ | $\mathcal{L}_{contra}$ | $\mathcal{L}_{consist}$ | random | diffusion | Yale | | | ORL | | |
|---|---|---|---|---|---|---|---|---|---|---|---|
| | | | | | | ACC | NMI | ARI | ACC | NMI | ARI |
| (a) | ✓ | | | | | 0.260 | 0.306 | 0.058 | 0.923 | 0.963 | 0.892 |
| (b) | ✓ | ✓ | | | | 0.670 | 0.708 | 0.517 | 0.930 | 0.960 | 0.893 |
| (c) | ✓ | ✓ | | ✓ | | 0.684 | 0.704 | 0.514 | 0.925 | 0.961 | 0.895 |
| (d) | ✓ | ✓ | | ✓ | ✓ | 0.712 | 0.743 | 0.568 | 0.930 | 0.959 | 0.885 |
| (e) | ✓ | | ✓ | | | 0.667 | 0.717 | 0.533 | 0.925 | 0.964 | 0.896 |
| (f) | ✓ | | ✓ | | ✓ | 0.694 | 0.744 | 0.567 | 0.933 | 0.964 | 0.904 |
| (g) | ✓ | ✓ | ✓ | ✓ | | 0.687 | 0.716 | 0.526 | 0.931 | 0.964 | 0.897 |
| (h) | ✓ | ✓ | ✓ | ✓ | ✓ | **0.716** | **0.751** | **0.584** | **0.943** | **0.966** | **0.911** |

Table 6: Ablation study on BBC-Sport and Scene-15 dataset

| | $\mathcal{L}_{rec}$ | $\mathcal{L}_{contra}$ | $\mathcal{L}_{consist}$ | random | diffusion | BBC-Sport | | | Scene-15 | | |
|---|---|---|---|---|---|---|---|---|---|---|---|
| | | | | | | ACC | NMI | ARI | ACC | NMI | ARI |
| (a) | ✓ | | | | | 0.778 | 0.786 | 0.682 | 0.481 | 0.493 | 0.312 |
| (b) | ✓ | ✓ | | | | 0.695 | 0.776 | 0.648 | 0.492 | 0.495 | 0.331 |
| (c) | ✓ | ✓ | | ✓ | | 0.701 | 0.764 | 0.590 | 0.493 | 0.498 | 0.327 |
| (d) | ✓ | ✓ | | ✓ | ✓ | 0.959 | 0.871 | 0.912 | 0.493 | 0.494 | 0.333 |
| (e) | ✓ | | ✓ | | | 0.432 | 0.200 | 0.107 | 0.477 | 0.493 | 0.321 |
| (f) | ✓ | | ✓ | | ✓ | 0.416 | 0.179 | 0.099 | 0.493 | 0.503 | 0.330 |
| (g) | ✓ | ✓ | ✓ | ✓ | | 0.681 | 0.712 | 0.558 | 0.490 | 0.496 | 0.330 |
| (h) | ✓ | ✓ | ✓ | ✓ | ✓ | **0.965** | **0.885** | **0.912** | **0.498** | **0.504** | **0.339** |

Table 7: Ablation study on MSRC-v1 and Reuters dataset

| | $\mathcal{L}_{rec}$ | $\mathcal{L}_{contra}$ | $\mathcal{L}_{consist}$ | random | diffusion | MSRC-v1 | | | Reuters | | |
|---|---|---|---|---|---|---|---|---|---|---|---|
| | | | | | | ACC | NMI | ARI | ACC | NMI | ARI |
| (a) | ✓ | | | | | 0.770 | 0.758 | 0.662 | 0.502 | 0.347 | 0.267 |
| (b) | ✓ | ✓ | | | | 0.795 | 0.780 | 0.699 | 0.510 | 0.347 | 0.269 |
| (c) | ✓ | ✓ | | ✓ | | 0.800 | 0.781 | 0.700 | 0.535 | 0.364 | 0.286 |
| (d) | ✓ | ✓ | | ✓ | ✓ | 0.893 | 0.845 | 0.902 | 0.507 | 0.344 | 0.258 |
| (e) | ✓ | | ✓ | | | 0.781 | 0.749 | 0.660 | 0.542 | 0.360 | 0.279 |
| (f) | ✓ | | ✓ | | ✓ | 0.790 | 0.770 | 0.681 | 0.546 | 0.360 | 0.270 |
| (g) | ✓ | ✓ | ✓ | ✓ | | 0.863 | 0.818 | 0.748 | 0.551 | 0.340 | 0.261 |
| (h) | ✓ | ✓ | ✓ | ✓ | ✓ | **0.950** | **0.908** | **0.887** | **0.587** | **0.376** | **0.287** |

Table 8: Ablation study on LandUse-21 and Caltech101-20 dataset

| | $\mathcal{L}_{rec}$ | $\mathcal{L}_{contra}$ | $\mathcal{L}_{consist}$ | random | diffusion | LandUse-21 | | | Caltech101-20 | | |
|---|---|---|---|---|---|---|---|---|---|---|---|
| | | | | | | ACC | NMI | ARI | ACC | NMI | ARI |
| (a) | ✓ | | | | | 0.287 | 0.329 | 0.166 | 0.586 | 0.719 | 0.494 |
| (b) | ✓ | ✓ | | | | 0.302 | 0.352 | 0.151 | 0.607 | 0.713 | 0.527 |
| (c) | ✓ | ✓ | | ✓ | | 0.317 | 0.349 | 0.160 | 0.632 | 0.738 | 0.544 |
| (d) | ✓ | ✓ | | ✓ | ✓ | 0.289 | 0.330 | 0.136 | 0.639 | 0.724 | 0.552 |
| (e) | ✓ | | ✓ | | | 0.324 | 0.345 | 0.160 | 0.552 | 0.687 | 0.460 |
| (f) | ✓ | | ✓ | | ✓ | 0.316 | 0.361 | 0.157 | 0.587 | 0.715 | 0.477 |
| (g) | ✓ | ✓ | ✓ | ✓ | | 0.292 | 0.336 | 0.139 | 0.628 | 0.722 | 0.552 |
| (h) | ✓ | ✓ | ✓ | ✓ | ✓ | **0.327** | **0.352** | **0.163** | **0.641** | **0.733** | **0.554** |

Table 9: Ablation study on ALOI-100 and STL10 dataset

| | $\mathcal{L}_{rec}$ | $\mathcal{L}_{contra}$ | $\mathcal{L}_{consist}$ | random | diffusion | ALOI-100 | | | STL10 | | |
|---|---|---|---|---|---|---|---|---|---|---|---|
| | | | | | | ACC | NMI | ARI | ACC | NMI | ARI |
| (a) | ✓ | | | | | 0.827 | 0.893 | 0.745 | 0.901 | 0.832 | 0.776 |
| (b) | ✓ | ✓ | | | | 0.856 | 0.894 | 0.759 | 0.550 | 0.578 | 0.391 |
| (c) | ✓ | ✓ | | ✓ | | 0.872 | 0.906 | 0.789 | 0.841 | 0.815 | 0.759 |
| (d) | ✓ | ✓ | | ✓ | ✓ | 0.868 | 0.902 | 0.776 | 0.866 | 0.820 | 0.708 |
| (e) | ✓ | | ✓ | | | 0.868 | 0.901 | 0.774 | 0.702 | 0.753 | 0.548 |
| (f) | ✓ | | ✓ | | ✓ | 0.869 | 0.900 | 0.776 | 0.618 | 0.582 | 0.417 |
| (g) | ✓ | ✓ | ✓ | ✓ | | 0.871 | 0.901 | 0.797 | 0.781 | 0.795 | 0.666 |
| (h) | ✓ | ✓ | ✓ | ✓ | ✓ | **0.891** | **0.911** | **0.805** | **0.968** | **0.920** | **0.930** |

Table 10: Ablation study on HandWritten and MNIST-3V dataset

| | $\mathcal{L}_{rec}$ | $\mathcal{L}_{contra}$ | $\mathcal{L}_{consist}$ | random | diffusion | HandWritten | | | MNIST-3V | | |
|---|---|---|---|---|---|---|---|---|---|---|---|
| | | | | | | ACC | NMI | ARI | ACC | NMI | ARI |
| (a) | ✓ | | | | | 0.967 | 0.926 | 0.926 | 0.949 | 0.928 | 0.898 |
| (b) | ✓ | ✓ | | | | 0.971 | 0.935 | 0.935 | 0.988 | 0.966 | 0.974 |
| (c) | ✓ | ✓ | | ✓ | | 0.974 | 0.940 | 0.942 | 0.986 | 0.961 | 0.970 |
| (d) | ✓ | ✓ | | ✓ | ✓ | 0.977 | 0.946 | 0.948 | 0.988 | 0.964 | 0.973 |
| (e) | ✓ | | ✓ | | | 0.970 | 0.933 | 0.933 | 0.956 | 0.932 | 0.909 |
| (f) | ✓ | | ✓ | | ✓ | 0.977 | 0.944 | 0.948 | 0.988 | 0.964 | 0.973 |
| (g) | ✓ | ✓ | ✓ | ✓ | | 0.970 | 0.934 | 0.934 | 0.988 | 0.965 | 0.973 |
| (h) | ✓ | ✓ | ✓ | ✓ | ✓ | **0.978** | **0.948** | **0.951** | **0.989** | **0.966** | **0.975** |

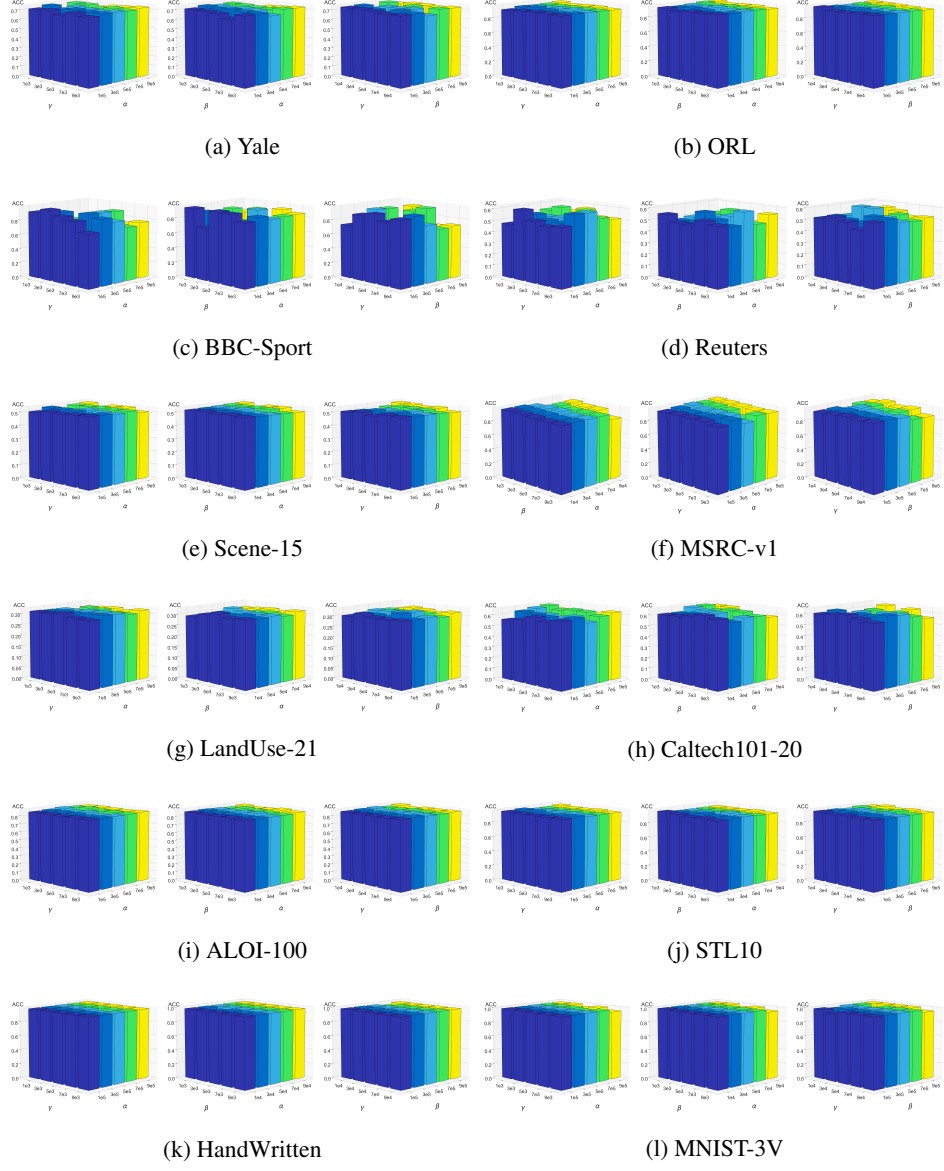

(a) Yale

(b) ORL

(c) BBC-Sport

(d) Reuters

(e) Scene-15

(f) MSRC-v1

(g) LandUse-21

(h) Caltech101-20

(i) ALOI-100

(j) STL10

(k) HandWritten

(l) MNIST-3V

Figure 4: Sensitivity Analysis of the MANGO Model to Parameters $\alpha$, $\beta$ and $\gamma$ on Twelve Datasets

### A.3 The Use of Large Language Models

During this research and the writing of this paper, we incorporated a Large Language Model (LLM) as an auxiliary tool to improve text processing efficiency and facilitate preliminary literature search preparation. It should be clarified that this tool's use was strictly limited to auxiliary tasks and did not participate in the core aspects of this research, including but not limited to research design, primary data collection, experimental workflow, statistical analysis, and the derivation and demonstration of scientific conclusions. The scientific integrity, rigor, and originality of the core research content are the sole responsibility of the authors.

Specifically, the LLM's auxiliary role in this research focused on the following two aspects:

(1)Text Polishing and Grammar Standardization: Optimizing the language expression of selected paragraphs in the first draft of the paper primarily involved correcting grammatical errors, improving sentence fluency, and assisting with standardizing academic terminology to ensure that the text adheres to the language logic and formatting requirements of academic writing. The final text's academic content, logical structure, and core ideas were all reviewed and confirmed by the authors.

(2)Preliminary Literature Review Assistance: During the literature search phase, LLM assisted in generating a preliminary conceptual framework and keyword list for a specific research field, providing reference for the authors to determine the scope of their literature search and select their search strategy. It should be emphasized that all the literature included in the literature review of this study were read in full by the authors one by one, and their research relevance, content accuracy and academic value were independently verified before final determination. The literature recommendation results generated by the model were not directly used.

