# OpenReview forum: "Multi-Scale Diffusion-Guided Graph Learning with Power-Smoothing Random Walk Contrast for Multi-View Clustering"
_ICLR.cc/2026/Conference — ICLR 2026 Poster_

### Official Review · Reviewer_1JZN · 2025-10-29

**Soundness:** 3
**Presentation:** 3
**Contribution:** 3
**Rating:** 6
**Confidence:** 5

**Summary:**

This work tackles several challenges in graph-based deep multi-view clustering, such as limited representation from static graphs, the influence of false negatives in contrastive learning, and the balance between cross-view consistency and view-specific discrimination. To address these, the authors introduce MANGO, a unified framework that combines adaptive multi-scale diffusion, random walk-based contrastive learning, and structure-aware view consistency modeling, and validate its effectiveness on multiple datasets.

**Strengths:**

1.	The model is well-motivated and effectively leverages rich information from multi-view data by combining adaptive multi-scale diffusion, random walk-based contrastive learning, and structure-aware view consistency modeling, leading to improved clustering performance.
2.	The workflow is clearly illustrated, accompanied by precise explanations.
3.	The baselines used for comparison are representative, covering both deep learning-based and shallow methods.

**Weaknesses:**

1. The power operation in the power-smoothing-induced contrastive learning is not described in detail, and the underlying mechanism that gives it an advantage over the standard InfoNCE loss is unclear.
2. The relationships between different modules are not well explained and feel somewhat disjointed, for example, the connection and distinction between component A in the Graph Refinement module and the random walk-based correction mechanism.
3. The experimental section does not provide the sources of the datasets used.
4. The analysis of Figure 2 (t-SNE visualization) in the paper is rather superficial and lacks in-depth discussion.

**Questions:**

On the Reuters dataset, the proposed method outperforms other baselines in terms of ACC and NMI, but performs significantly worse than the CVCL model in ARI. Could the authors clarify the reason for this discrepancy?

---

> ### Author Response · Authors · 2025-11-21
>
> We sincerely thank the reviewers for recognizing our work. The weaknesses you pointed out were very pertinent and insightful, providing direction for further improvement of the paper. Below are detailed responses and revision plans for each weakness.
>
> ### Weakness 1: Insufficient detail on power-smoothing mechanism
> Thank you very much for pointing this out. We agree that the previous description was too brief. The core motivation for introducing the β-power operation is to non-linearly smooth the distribution of the negative sample terms to suppress the influence of potentially extreme noise samples (i.e., "false negatives").
> In the standard InfoNCE loss, a highly similar "false negative" sample can dominate the negative sample terms, generating a huge gradient that pushes semantically similar samples away, thus harming model performance. Our β-power operation operates on the entire negative sample summation term. It reduces the relative gap between larger values ​​while increasing the importance of smaller values. In this way, even if there are one or two highly similar outliers in the negative sample set, their contribution to the overall negative sample term is relatively weakened after the β-power, thereby reducing the harmful gradients brought by these noisy samples.
>
>
> ### Weakness 2: Disjointed module relationships
> This is a very valuable suggestion, pointing out a key deficiency in the "narrative coherence" of our paper. We will focus on strengthening this section in the revised version.
>
> The random walk correction aims to serve contrastive learning. It utilizes the t-step transition matrix T to identify and reduce the weights of "false negative" samples, thereby purifying the training signal for contrastive learning. Multi-scale diffusion is used to generate the consensus affinity matrix for the final spectral clustering; their functions are complementary and executed sequentially.
>
>
> ### Weakness 3: Missing dataset sources
> We sincerely apologize for this oversight. It did indeed affect the reproducibility of our work.
> In fact, all datasets used are publicly available benchmarks in multi-view clustering. In addition to the statistical information already presented in the main text, we have now included the corresponding references for each dataset to enhance clarity and transparency: Yale [1], ORL [2], BBCSport [1], Reuters [2], Scene15 [3], MSRCv1 [2], LandUse-21 [4], Caltech101-20 [5], ALOI-100 [6], STL-10 [7], Handwritten [8], and MNIST-3V [9]. Furthermore, the source code and several datasets were provided in the supplementary material upon submission to support reproducibility.
>
>
> ### Weakness 4: Superficial t-SNE analysis
> We completely agree with your point of view. The current analysis only stays at the level of "clearer and more compact clustering" and fails to connect with the core innovations of the model for in-depth interpretation. We will rewrite and deepen the analysis of Figure 2 in the revised version.
>
>
> [1] Chowdhury et al. (2025). Deep multi-view clustering: A survey of the contemporary techniques. Information Fusion.
>
> [2] Du et al. "Multiview subspace clustering with multilevel representations and adversarial regularization." IEEE TNNLS (2022).
>
> [3] Fei-Fei & Perona. "A bayesian hierarchical model for learning natural scene categories." CVPR 2005.
>
> [4] Zong et al. "Self-supervised deep multiview spectral clustering." IEEE TNNLS (2022).
>
> [5] Peng, Xi, et al. "COMIC: Multi-view clustering without parameter selection." International conference on machine learning. ICML, 2019.
>
> [6] Yuan et al. "Prototype matching learning for incomplete multi-view clustering." IEEE TIP (2025).
>
> [7] Yu et al. "A non-parametric graph clustering framework for multi-view data." AAAI 2024.
>
> [8] van Breukelen et al. "Handwritten digit recognition by combined classifiers." Kybernetika 1998.
>
> [9] Zhou et al. "Multi-view spectral clustering with optimal neighborhood Laplacian matrix." AAAI 2020.

---

### Official Review · Reviewer_oBPZ · 2025-10-29

**Soundness:** 2
**Presentation:** 3
**Contribution:** 3
**Rating:** 4
**Confidence:** 4

**Summary:**

The manuscript introduces a MVC framework built on graph structures, aiming to address limitations in existing approaches. The framework combines three core modules: a multi-scale diffusion process guided by local entropy to integrate similarity matrices and capture both local and global semantics; a random walk–based correction mechanism that reduces the impact of false negatives via a $\beta$-power reweighting scheme; and a structure-aware view consistency module that aligns embeddings across views while preserving view-specific features. Experiments on 12 benchmark datasets indicate that the proposed method outperforms all baseline approaches.

**Strengths:**

1. The proposed method employs an interesting multi-scale diffusion mechanism to overcome the performance bottleneck caused by fixed diffusion step sizes. It dynamically fuses similarity information across different steps while considering both local structure exploration and global semantic modeling, enabling effective capture of multi-granularity structural information.
2. The experimental section includes not only quantitative performance comparisons but also visualizations of graphs produced by different methods, offering deeper insights.
3. The code is provided by the authors in the supplementary materials, enabling reproducibility.

**Weaknesses:**

1. It is unclear how the view-aware attention mechanism mentioned in the Introduction, as part of the structure-aware cross-view contrastive learning mechanism, is implemented in the Method section. The authors should clarify this.
2. The uniform weight matrix W appearing in Equation (5) lacks a clear definition. The authors should explicitly describe how this matrix is formulated or obtained.
3. The graph refinement process involves T-step diffusion, but the effect of different T values on model performance is not discussed. It is also unclear whether the same T value is applied across different datasets.
4. Several formatting issues are noted in the manuscript. In particular, some references lack page number information.

**Questions:**

See weaknesses.

---

> ### Author Response · Authors · 2025-11-21
>
> Thank you very much for your recognition and in-depth feedback on this research. Below are our point-by-point responses to the concerns you raised.
>
> ### Weakness 1: Implementation of View-Aware Attention Mechanism
> We appreciate the reviewers' keen observation. We must apologize for this inconsistency in wording. In the introduction, we used the term "view-aware attention mechanism" to describe our conceptual goal—to promote consistency while maintaining view-specificity. However, in the concrete implementation, we achieve this goal through a more direct and efficient bidirectional mapping mechanism (Equations 7-9), which explicitly learns mapping functions between views to align semantics while preserving the original view-specific embeddings.
>
> ### Weakness 2: Definition of Uniform Weight Matrix W
> This is a very important correction, and we apologize for omitting this definition in the main text. In the cross-view contrast loss of Equation (5), the matrix $W$ is initialized as a uniform weight matrix, the intention of which is to assign the same weight to all samples j from other views without prior knowledge, i.e.,$W_{ij}=\frac{1}{n-1}$​ (for all j≠i).
>
>
>
> ### Weakness 3: Unclear Impact of Different T Values on Model Performance
> Thank you for raising this crucial question about the hyperparameter T. We agree that discussing the impact of T is essential for understanding model behavior. In all reported experiments, we consistently set the maximum diffusion steps T to 3. This value was chosen based on preliminary experiments and strikes a good balance between computational cost and capturing long-range dependencies. We will add a new section discussing the impact of different T values ​​on performance and supplement it with relevant charts showing the trends of ACC under different T values.
> | Dataset | T=1 | T=2 | T=3 | T=4 | T=5 |
> | ------- | --- | --- | --- | --- | --- |
> | Yale    |0.701|0.698|0.716|0.704|0.710|
> | ORL     |0.927|0.931|0.943|0.935|0.932|
> | BBCSport|0.894|0.920|0.965|0.966|0.953|
> | Reuters |0.529|0.545|0.587|0.566|0.547|
> | Scene-15|0.491|0.494|0.498|0.502|0.491|
> | MSRC-v1 |0.900|0.917|0.950|0.925|0.913|
> |LandUse-21|0.296|0.308|0.327|0.328|0.347|
> |Caltech101_20|0.589|0.593|0.641|0.636|0.608|
> |ALOI_100 |0.876|0.896|0.891|0.856|0.856|
> | STL10   |0.968|0.956|0.968|0.954|0.955|
> |HandWritten|0.977|0.978|0.978|0.978|0.976|
> | MNIST   |0.984|0.986|0.989|0.990|0.992|
>
> The results show that T=3 is the optimal number of diffusion steps for generalization performance. In most cases, when T=1, the model only fuses a single diffusion matrix, resulting in insufficient global semantic capture and a low ACC. As T increases to 3, the model performance gradually improves. However, when the T value is too large, excessive diffusion steps dilute local structural information, leading to a decrease in model performance. Furthermore, a higher T value also implies higher computational cost and time. Considering all factors, we decided to set T to 3.
>
> ### Weakness 4: Formatting issues with references
> We greatly appreciate your meticulous checking of the formatting issues. We acknowledge this was an oversight in integrating a large number of references. We will thoroughly review all references in the manuscript and revise them strictly according to the ICLR conference formatting requirements.

---

### Official Review · Reviewer_FL2j · 2025-10-29

**Soundness:** 3
**Presentation:** 2
**Contribution:** 3
**Rating:** 4
**Confidence:** 5

**Summary:**

This paper proposes MANGO, a deep multi-view clustering framework that integrates random walk–based contrastive learning, view consistency alignment, and adaptive diffusion to learn discriminative and robust multi-view embeddings. The authors validate the effectiveness of the proposed method through extensive experiments, including performance comparisons, ablation studies, and sensitivity analyses.

**Strengths:**

The framework effectively combines improved contrastive learning, view consistency, and adaptive diffusion in a unified architecture, providing a coherent approach to multi-view clustering, and the experimental evaluation across datasets of varying types and scales provides strong evidence for the effectiveness of the proposed method.

**Weaknesses:**

1. The variables in the flowchart are not clearly labeled; many matrices and components lack explanation.
2. Formula (6) involves a balancing parameter, but the strategy for choosing its value is not described.
3. The manuscript claims that the multi-scale diffusion mechanism jointly models fine-grained local structures and overall global semantics; however, it is unclear whether any evidence is provided to support this claim. Further clarification or validation is needed.
4. The description of the consistency loss (Formula 9) is unclear, making it difficult to understand.
5. In the related work section, deep multi-view clustering methods are categorized into joint methods, alignment-based methods, and other methods, but the basis for this classification is not explained.

**Questions:**

See Weakness.

---

> ### Author Response · Authors · 2025-11-21
>
> We greatly appreciate the reviewers' constructive comments on our paper. Below, we have provided point-to-point responses to the issues you raised.
>
> ### Weakness 1: Unclear Variable Labeling in the Flowchart
> We thank the reviewer for pointing out this important issue. We acknowledge that Figure 1 in the original manuscript did not have sufficient symbol annotation. In the revised version, we will take the following steps to improve this: Add more detailed legends within the figures to explain the inputs, outputs, and meanings of each key module. In the descriptive text surrounding Figure 1, explain the roles of each key variable and component, ensuring a one-to-one correspondence with the formulas and descriptions in the main text.
>
> ### Weakness 2: Strategy for Selecting the Balancing Parameter in Formula (6)
> We acknowledge that this important detail is indeed missing from the original text. Formula (6) is the total contrast learning loss $L_{contra}= L_{intra}+ \mu L_{inter}$​, where $\mu$ is a hyperparameter balancing the in-view contrast loss and the inter-view contrast loss.Specifically, μ is selected from the set {0.01, 0.1, 1, 10} through a grid search and is ultimately set to 0.1 across all datasets. We will elaborate on the search range and final selected value of this parameter in Section 4.1 of the revised version.
>
>
>
> ### Weakness 3: Insufficient Evidence for Multi-Scale Diffusion Mechanism
> We greatly appreciate the insightful question raised by the reviewer. We fully agree that a mere description of the method is insufficient to provide convincing evidence; empirical evidence is needed to support the assertion that "multi-scale diffusion can jointly model local and global structures." Based on your suggestion, we conducted in-depth ablation experiments, and the results strongly validate the effectiveness of this mechanism. Specifically:
>
> First, we clarify the theoretical motivation: fixed-step diffusion faces a fundamental trade-off—small step sizes primarily capture local neighborhood structures but struggle to infer long-range semantic relationships; while large step sizes can explore global semantic communities through information propagation but may lead to over-smoothing of local details. The core advantage of our entropy-guided multi-scale fusion mechanism lies in its ability to adaptively fuse structural information under different step sizes, thereby simultaneously utilizing the accuracy of local structures and the robustness of global communities.
>
> We have supplemented your question with experimental evidence. We conducted systematic ablation experiments, comparing the complete MANGO model with a variant (denoted as MANGO-w) that removes multi-scale fusion and uses only a fixed 3-step diffusion step size.
>
> The experimental results are as follows:
> | Dataset | Method | ACC | NMI | ARI |
> | --------- | ------ | ------------ | --------- | ---------- |
> | Yale          | MANGO-w| 0.715   | 0.750  | 0.572   |
> |               | MANGO  | 0.729   | 0.757  | 0.582   |
> | ORL           | MANGO-w| 0.928   | 0.961  | 0.895   |
> |               | MANGO  | 0.926   | 0.961  | 0.895   |
> | BBC-Sport     | MANGO-w| 0.953   | 0.906  | 0.921   |
> |               | MANGO  | 0.959   | 0.907  | 0.932   |
> | Reuters       | MANGO-w| 0.499   | 0.321  | 0.253   |
> |               | MANGO  | 0.583   | 0.351  | 0.281   |
> | Scene-15      | MANGO-w| 0.496   | 0.502  | 0.343   |
> |               | MANGO  | 0.497   | 0.499  | 0.388   |
> | MSRC-v1       | MANGO-w| 0.929   | 0.875  | 0.841   |
> |               | MANGO  | 0.949   | 0.904  | 0.884   |
> | LandUse21     | MANGO-w| 0.301   | 0.344  | 0.148   |
> |               | MANGO  | 0.315   | 0.344  | 0.160   |
> | Caltech101-20 | MANGO-w| 0.597   | 0.713  | 0.494   |
> |               | MANGO  | 0.619   | 0.725  | 0.535   |
> | ALOI-100      | MANGO-w| 0.828   | 0.878  | 0.731   |
> |               | MANGO  | 0.887   | 0.910  | 0.803   |
> | STL-10        | MANGO-w| 0.926   | 0.862  | 0.837   |
> |               | MANGO  | 0.960   | 0.911  | 0.912   |
> | HandWritten   | MANGO-w| 0.973   | 0.944  | 0.944   |
> |               | MANGO  | 0.978   | 0.948  | 0.952   |
> | MNIST-3V      | MANGO-w| 0.985   | 0.960  | 0.970   |
> |               | MANGO  | 0.987   | 0.962  | 0.972   |
>
> The results show that for large-scale data with complex structures, relying solely on structural information at a single scale is insufficient. The model must be able to simultaneously perceive subtle local relationships and macroscopic global communities, which is precisely what multi-scale diffusion provides.

---

> > ### Author Response · Authors · 2025-11-21
> >
> > ### Weakness 4: Unclear Description of Consistency Loss
> > We appreciate you pointing out this issue. The core objective of the view consistency module is to learn a bidirectional mapping across views, ensuring that representations of the same category in different views are as close as possible in the shared semantic space. Specifically, this involves using a mapping function $f_{p\rightarrow q}$ to transform the embedding $Z^p$ of view $p$ into $\hat Z^p$, making $\hat Z^p$ as close as possible to the original embedding $Z^q$ of view $q$, thus achieving cross-view semantic alignment. By minimizing the distance between the mapped representations, we ensure that the subsequent fusion module receives aligned embeddings, rather than conflicting noise.
> >
> >
> >
> > ### Weakness 5: Unexplained Basis for Related Work Classification
> >
> > This point is crucial. In the revised version, we will supplement the basis for this classification, explaining that it is based on how the model processes and integrates information from multiple views:
> >
> > Joint methods: These aim to learn the representations and clustering structures of all views simultaneously through a unified model or objective function, emphasizing collaboration between views.
> >
> > Alignment-based methods: These primarily focus on mapping the representations of different views to a shared subspace and enforcing consistency within that subspace.
> >
> > Other methods: These include mainstream methods that are difficult to categorize into the above two types, such as methods specifically designed to handle incomplete views or noise.

---

> ### Comment · Reviewer_FL2j · 2025-11-24
> **Comment on 14963**
>
> Thanks for the author's detailed response, they have resolved all of my issues. Therefore, I have decided to change my rating to show support.

---

> > ### Author Response · Authors · 2025-11-26
> >
> > We are delighted to receive your positive feedback and are pleased to know that our responses have addressed all your concerns. Thank you for your valuable insights throughout the review process, which have been instrumental in improving our work. We sincerely appreciate your updated rating and supportive decision.

---

### Official Review · Reviewer_tTks · 2025-10-30

**Soundness:** 3
**Presentation:** 3
**Contribution:** 3
**Rating:** 6
**Confidence:** 5

**Summary:**

This paper proposes a novel deep multi-view clustering framework named MANGO, which learns a discriminative multi-view embedding by integrating three key components: a random walk–modified contrastive learning module, a view consistency module, and an adaptive diffusion module. Specifically, the contrastive module dynamically adjusts the weights of negative samples to better handle complex data distributions; the view consistency module achieves deep alignment across feature spaces via a bidirectional projection mechanism; and the adaptive diffusion module captures multi-scale structural information while mitigating over-smoothing and information loss. Extensive experiments demonstrate the effectiveness of the proposed MANGO.

**Strengths:**

1.The paper presents a well-structured deep multi-view clustering framework that integrates random walk–modified contrastive learning, view consistency, and adaptive diffusion modules, each complementing the others effectively.
2.Extensive experiments conducted on 12 multi-view datasets of varying scales and types convincingly demonstrate the superiority and robustness of the proposed method.
3.The implementation details of the experiments are presented in a detailed manner.

**Weaknesses:**

1. The manuscript does not clearly explain the physical meaning of the transfer matrix $M$ in the random walk-based correction mechanism. Its connection with the similarity matrix and its role in the correction process require further clarification.
2. The rationale behind the hybrid regularization in Equation (2) is insufficiently detailed. The authors should clarify how this regularization helps prevent overfitting and improves model generalization.
3. The manuscript does not indicate the absence of standard deviations in Table 2, nor does it clarify whether the reported results are based on a single run or the average performance over multiple runs.
4. The indexing of equations in the pseudocode is not consistent or properly formatted.

**Questions:**

See Weakness section.

---

> ### Author Response · Authors · 2025-11-21
>
> We are grateful to the Reviewer for the insightful feedback. Our responses to the comments are as follows:
>
> ### Weakness 1: Physical Meaning of the Transfer Matrix
> The transfer matrix $M^t$ is derived from the tt-step random walk on the graph, where each entry $M^t_{ij}t$​ represents the probability of transitioning from node $i$ to node $j$ after tt steps. This matrix captures multi-hop semantic relationships between samples, which helps in identifying semantically similar pairs that are not directly connected in the original similarity graph. By integrating $M^t$ into the contrastive learning framework, we can better distinguish true negative pairs from false ones. We will clarify this in the revised manuscript by adding a more detailed explanation in Section 3.2.
> ### Weakness 2:Rationale Behind Hybrid Regularization in Equation
> Thank you for pointing this out. The hybrid regularization term combines $l_1$-norm and Frobenius norm.
> The $l_1$​-norm promotes sparsity in the coefficient matrix $C^v$, which helps in selecting a subset of meaningful connections and reducing overfitting. The Frobenius norm stabilizes the solution and prevents extreme values, enhancing generalization. The balance parameter $\lambda$ controls the trade-off between sparsity and stability. We will add a brief explanation in Section 3.1 to justify the design and its role in improving model robustness.
>
> ### Weakness 3: Standard Deviations and Result Reproducibility in Table 2
> We sincerely apologize for the previous oversight. All results in Table 2 are the average of 10 independent runs. We have calculated the standard deviation, but it is not listed due to space limitations. We will add ±std in the revised Table 2 and explain it in the figure captions.The following are comparison results with some of the latest methods.
> | Dataset       | Method | ACC            | NMI           | ARI           |
> | ------------- | ------ | -------------- | ------------- | ------------- |
> | Yale          | CANDY  | 0.590 ± 0.025  | 0.600 ± 0.029 | 0.353 ± 0.035 |
> |               | SCM    | 0.574 ± 0.027  | 0.586 ± 0.020 | 0.395 ± 0.026 |
> |               | MANGO  | 0.729 ± 0.013  | 0.757 ± 0.011 | 0.582 ± 0.015 |
> | ORL           | CANDY  | 0.672 ± 0.019  | 0.323 ± 0.008 | 0.567 ± 0.020 |
> |               | SCM    | 0.673 ± 0.030  | 0.821 ± 0.017 | 0.558 ± 0.036 |
> |               | MANGO  | 0.926 ± 0.007  | 0.961 ± 0.004 | 0.895 ± 0.011 |
> | BBC-Sport     | CANDY  | 0.639 ± 0.0656 | 0.404 ± 0.053 | 0.358 ± 0.072 |
> |               | SCM    | 0.720 ± 0.034  | 0.565 ± 0.025 | 0.513 ± 0.031 |
> |               | MANGO  | 0.959 ± 0.012  | 0.907 ± 0.019 | 0.932 ± 0.018 |
> | Reuters       | CANDY  | 0.539 ± 0.024  | 0.284 ± 0.022 | 0.325 ± 0.017 |
> |               | SCM    | 0.459 ± 0.061  | 0.242 ± 0.040 | 0.201 ± 0.055 |
> |               | MANGO  | 0.583 ± 0.026  | 0.351 ± 0.013 | 0.281 ± 0.013 |
> | Scene-15      | CANDY  | 0.365 ± 0.022  | 0.359 ± 0.011 | 0.195 ± 0.013 |
> |               | SCM    | 0.325 ± 0.012  | 0.263 ± 0.014 | 0.150 ± 0.009 |
> |               | MANGO  | 0.497 ± 0.023  | 0.499 ± 0.004 | 0.388 ± 0.013 |
> | MSRC-v1       | CANDY  | 0.494 ± 0.029  | 0.287 ± 0.030 | 0.160 ± 0.034 |
> |               | SCM    | 0.665 ± 0.032  | 0.592 ± 0.034 | 0.493 ± 0.040 |
> |               | MANGO  | 0.949 ± 0.006  | 0.904 ± 0.013 | 0.884 ± 0.014 |
> | LandUse21     | CANDY  | 0.229 ± 0.027  | 0.258 ± 0.032 | 0.091 ± 0.019 |
> |               | SCM    | 0.260 ± 0.010  | 0.291 ± 0.010 | 0.278 ± 0.007 |
> |               | MANGO  | 0.315 ± 0.009  | 0.344 ± 0.003 | 0.160 ± 0.004 |
> | Caltech101-20 | CANDY  | 0.421 ± 0.040  | 0.393 ± 0.237 | 0.246 ± 0.019 |
> |               | SCM    | 0.518 ± 0.058  | 0.513 ± 0.020 | 0.627 ± 0.095 |
> |               | MANGO  | 0.619 ± 0.023  | 0.725 ± 0.014 | 0.535 ± 0.032 |
> | ALOI-100      | CANDY  | 0.696 ± 0.027  | 0.702 ± 0.210 | 0.568 ± 0.017 |
> |               | SCM    | 0.687 ± 0.015  | 0.800 ± 0.006 | 0.516 ± 0.013 |
> |               | MANGO  | 0.887 ± 0.009  | 0.910 ± 0.003 | 0.803 ± 0.009 |
> | STL-10        | CANDY  | 0.677 ± 0.008  | 0.615 ± 0.010 | 0.594 ± 0.009 |
> |               | SCM    | 0.937 ± 0.016  | 0.860 ± 0.021 | 0.872 ± 0.011 |
> |               | MANGO  | 0.960 ± 0.013  | 0.911 ± 0.012 | 0.912 ± 0.021 |
> | HandWritten   | CANDY  | 0.951 ± 0.016  | 0.885 ± 0.021 | 0.842 ± 0.032 |
> |               | SCM    | 0.730 ± 0.055  | 0.697 ± 0.030 | 0.600 ± 0.052 |
> |               | MANGO  | 0.978 ± 0.001  | 0.948 ± 0.001 | 0.952 ± 0.001 |
> | MNIST-3V      | CANDY  | 0.994 ± 0.002  | 0.964 ± 0.003 | 0.971 ± 0.002 |
> |               | SCM    | 0.931 ± 0.027  | 0.862 ± 0.015 | 0.830 ± 0.024 |
> |               | MANGO  | 0.987 ± 0.000  | 0.962 ± 0.001 | 0.972 ± 0.000 |
> ### Weakness 4: Inconsistent Equation Indexing in the Pseudocode
> We acknowledge this formatting issue. In the revised version, we will ensure that all equation references in Algorithm 1 are consistently formatted and correctly linked to their respective equations in the main text.

---

### Author Response · Authors · 2025-12-01

Dear ICLR 2026 Conference PC, SAC, AC, and Reviewers,​

We sincerely thank the PC, SAC, and AC of the ICLR community for their dedicated efforts during the review process, and we greatly appreciate the four reviewers for their recognition and constructive feedback.​

All reviewers acknowledged the novelty and effectiveness of our method (**Reviewer tTks:** a well-structured deep multi-view clustering framework; **Reviewer FL2j:**  providing a coherent approach to multi-view clustering; **Reviewer oBPZ:** The proposed method employs an interesting multi-scale diffusion mechanism to overcome the performance bottleneck caused by fixed diffusion step sizes; **Reviewer 1JZN:** The model is well-motivated and effectively leverages rich information from multi-view data by combining adaptive multi-scale diffusion, random walk-based contrastive learning, and structure-aware view consistency modeling”), as well as the comprehensiveness of our experimental evaluation.

We carefully addressed all reviewer concerns in our rebuttal, providing detailed explanations and supplementary experiments.​
Following our rebuttal, **Reviewer FL2j confirmed that their concerns were fully resolved, increasing their score from 4 to 8 on November 24**. Thus, with one reviewer updated, the scores currently stand at 8, 6, 6, and 4. **Reviewer oBPZ’s concerns (4 points) relate primarily to technical detail descriptions or minor presentation issues** (e.g., missing implementation details of the View-Aware Attention Mechanism, definition of the Uniform Weight Matrix W, impact of different T values, and reference formatting), which do not reflect flaws in the model design or experimental results and can be readily addressed. Concerns from the 6-point reviewers (tTks, 1JZN) have also been comprehensively clarified.​

**In summary, all four reviewers have recognized the novelty of our model and the validity of our experimental results. Following our rebuttal, Reviewer FL2j increased their score from 4 to 8, while the only remaining low-score reviewer, oBPZ (4 points), raised concerns that can be fully addressed through additional explanations, improved descriptions, and supplementary experiments. We have responded to each of these points in our rebuttal and completed all necessary revisions, and we believe these updates have fully resolved the reviewers’ concerns.**

We highly value the review and revision process and believe that the manuscript has been significantly improved in quality, technical transparency, and rigor. We respectfully ask the AC to consider our comprehensive responses and the substantive improvements in the final decision.​

Sincerely,

Authors of Submission #14963

---

### Meta-Review · Area_Chair_FvSm · 2026-01-08

**Summary:**

This paper introduce a unified framework named multi-scale diffusion-guided graph learning with power-smoothing random walk contrast (MANGO) for multi-view clustering by combining adaptive multi-scale diffusion, random walk-driven contrastive learning, and structure-aware view consistency modeling. The scores are 6, 8, 4 and 6 *(please see the Reviewer Score section)*. The strengths of this paper is on the incorporation of diffusion mechanism and good performance. Considering the comments from all reviewers, I tend to accept.

**Reviewer Concerns:**

The reviewer concerns are responded by the authors. Reviewer FL2j raised the score.

**Reviewer Scores:**

The current scores are 6, 4, 4 and 6. But the authors claimed that Reviewer FL2j raised the score from 4 to 8. I checked the revision history of Reviewer FL2j and found that he indeed raised score to 8 but the revision is not displayed. Also Reviewer FL2j has explicitly said to raise the score. So, I consider the final scores are 6, 8, 4, 6.

---

### Decision · Program_Chairs · 2026-01-26

Accept (Poster)